# Should the Glu Be Ten or Twenty? An Update on the Ongoing Debate on Gluten Safety Limits for Patients with Celiac Disease

**Inna Spector Cohen** [1] , **Andrew Day** [2] **and Ron Shaoul** [1,*]

1   Pediatric Gastroenterology & Nutrition institute, Ruth Children's Hospital of Haifa,
    Rambam Medical Center, Faculty of Medicine, Technion, Haifa 3109601, Israel; ejikrezin0vy@gmail.com
2   Department of Paediatrics, University of Otago Christchurch, Christchurch 8140, New Zealand;
    andrew.day@otago.ac.nz
*   Correspondence: ron.shaoul@gmail.com or r_shaoul@rambam.health.gov.il; Tel.: +972-50-2063333

**Abstract:** The only currently accepted and recommended treatment for individuals diagnosed with celiac disease (CD) is a strict life-long gluten-free diet (GFD). While the use of the GFD is well-established, strict adherence to diet is not easy to accomplish. In addition, the administration of a GFD may be compromised by inadvertent exposure to small amounts of gluten. International guidelines define a gluten-free product as one containing less than 20 parts per million (ppm), (20 milligrams of gluten per 1 kg of food) gluten. A number of reports have assessed the safe upper limit for gluten exposure for patients with CD, with general consensus that patients with CD should limit their daily intake to less than 50 mg.

**Keywords:** celiac disease; gluten; gluten free diet; guidelines

## 1. Introduction

Celiac disease (CD) is an immune-mediated small intestinal enteropathy triggered by the ingestion of gluten in genetically susceptible individuals possessing particular HLA genes. CD is present in around 1–2% of populations in Europe and Oceania [1–3], North and South America, North Africa, Middle East and India [1–5]. Recent reports indicate a higher prevalence in some populations, with rates of up to 3% in Denver, USA [6] and Sweden [7].

Gluten is a protein found in wheat and some related grains (such as rye and barley). These cereals are consumed in a typical western diet, with an average of 10–20 g per person [8] per day. Gluten is composed of prolamins (known as gliadins in wheat, hordein in barley and secalin in rye) and glutelins (glutenins in wheat) [9]. The prolamins are a complex group of alcohol-soluble proteins and constitute the major seed proteins in cereals. Overall, these proteins comprise around half of the proteins present in mature cereal grains. The prolamins are rich in proline and glutamine residues, making them resistant to digestion in the human gastrointestinal tract. In addition, these residues present in the undigested gluten fragments are excellent substrates for tissue transglutaminase (tTG) within the lamina propria [10]. Of the four described fractions of gliadin, the α-gliadin subunit, due to its high T-cell-stimulatory sequences, has the most significant effects while β, γ and ω subunits have milder toxicity [8,11,12].

The deamination of gliadin peptides by tTG generates a complex with high affinity for the DQ2 or DQ8 pockets present on antigen-presenting cells. This enables presentation of the peptides to some subsets of CD4 and CD8 T lymphocytes, prompting direct immune responses. This includes the release of pro-inflammatory cytokines (such as interferon-γ) and metalloproteinases. Together, these events lead to the typical histological changes present in the duodenal mucosa, characterized by villous

flattening, crypt hyperplasia, intraepithelial lymphocytosis, and increased cellularity (predominantly lymphocytes and plasmocytes) within the lamina propria [8].

## 2. Aspects of Gluten-Free Diet

To date, the only known effective treatment for CD is a life-long strict gluten-free diet (GFD). This dietary change involves the exclusion of any cereal containing gluten, including wheat, rye, barley and hybrids such as kamut and triticale [8].

The standard GFD prescription involves the use of foods that are naturally gluten-free (such as vegetables, legumes, naturally gluten-free cereals and fruits). Even if some products, especially processed ones, are labeled as "gluten-free", gluten is not always totally absent, as they could contain up to 20 parts per million (ppm) to bear such a claim, depending on the legal status of each country. Consequently, it is important to define the safe gluten amount for individuals with CD and to inform patients about food labels, in order to limit gluten ingestion.

In most patients, commencement of a GFD is followed by normalization of the histological changes in combination with resolution of serological and symptoms [9,13]. GFD also reduces mortality and the risk of complications associated with CD (such as malignancy or intestinal lymphoma) [9,14]. Patients who have ongoing gluten exposure have a two-to four-fold increased risk of non-Hodgkin's lymphoma, greater than a 30-fold increased risk of small intestinal adenocarcinoma and a 1.4-fold increased risk of death [15]. In addition, continuous gluten exposure, especially in the setting of untreated CD, is associated with an increased risk of other autoimmune diseases [16].

The important issue then is to define the safe gluten amount for individuals with CD [8]. Adherence to a GFD can be difficult because many processed foods contain gluten, including products that are contaminated with small amounts of gluten during processing [17]. Gluten can also be present as a component of pharmaceutical products [18] and religious ceremony foods [19].

Maintenance of a strict GFD impacts adversely on patients' daily activities, limiting social events such as eating out with consequent detrimental effects on quality of life [20]. Intentional or inadvertent exposure to gluten whilst on a GFD is the most common cause of non-response to a GFD [21]. Nonadherent patients may be asymptomatic and can have normal serology results, despite ongoing mucosal changes of villous atrophy and inflammation [22]. There remains controversy as to the safe amount of daily gluten consumption to avoid clinical consequences and mucosal damage in individuals with CD.

## 3. Relationships between Ongoing Exposure to Gluten and Outcomes in the Setting of a Gluten-Free Diet

Numerous scientific reports have demonstrated the positive outcomes and benefits of GFD in subjects with CD. These reports have described the short- and long-term outcomes of the maintenance of GFD in adults and children. Table 1 summarizes these studies.

The long-term outcomes of GFD were ascertained in a group of 390 (91 men and 299 women) adults followed for an average of more than 6 years after the diagnosis of CD [14]. The study group was on a GFD for at least 2 years. More than half (56.4%) of the subjects had persistent mucosal damage while one quarter of the group had positive anti-endomysial antibodies (EMA). The presence of intestinal damage was particularly associated with dietary non-adherence, which related in part to initial dietetic education. Overall, the laboratory and clinical findings were helpful in the identification of those individuals with ongoing intestinal damage.

A prospective gluten challenge study conducted in children with established CD showed clear correlations between the amount of gluten exposure and enteropathy [23]. This Swedish study involved 54 children who were allocated to receive 0.2 or 0.5 g/kg of gluten powder daily. Within four weeks most of the children in the lower gluten dose group and all of those receiving the higher dose had relapsed. All the remaining patients relapsed in the subsequent four weeks. The severity of mucosal

changes was greater in those who had received 0.5g/kg/day. There were no correlations between serological changes and histological severity.

Catassi et al. [24] also showed the dose-dependent effects of gluten exposure in children with CD. Children with known CD were given 100 or 500 mg gliadin/day (200 mg or 1 gr gluten/day, respectively) for four weeks. The children who received 100 mg daily did not report any symptoms and had minimal histological changes. In contrast, the children in the higher dose group had marked mucosal damage, even within this short period of exposure. Furthermore, three of these ten children had symptoms such as pale stools and loss of appetite. In a later study, this same group conducted a prospective, double-blind, placebo-controlled study involving 49 adult patients with CD [25]. Subjects were randomized to receive 10 mg of gluten daily, 50 mg of gluten daily or a placebo. The subjects given the higher dose of gluten for three months had significant mucosal damage, defined as worsening of the duodenal villous height/crypt depth (VH/CrD) ratio. Seven of the thirteen subjects in the low-dose gluten group also had a worsening of their VH/CrD ratio. Interestingly, there was no significant change in IgA anti-tissue transglutaminase antibodies and even a significant decrease in anti-gliadin (AGA) IgG antibodies was noted.

In contrast, another study did not demonstrate any relationship between villous atrophy and the extent of a GFD [26]. Thirty-nine adult patients receiving what the authors called a Codex GFD (based on the Codex Alimentarius Commission of the World Health Organization that allows up to 0.03% protein derived from gluten-containing grains to be included in so-called gluten-free foods, principally in the form of wheat starch and malt) and 50 patients consuming a non-gluten detectable-GFD (NGD-GFD) were compared. The presence of intra-epithelial lymphocytes or villous atrophy did not differ between groups. The authors concluded that persistent mucosal changes did not occur secondary to trace amounts of gluten.

In another study, Troncone et al. [27] assessed adherence to GFD in a group of adolescents who had been followed for many years after diagnosis of CD. Gluten exposure was assessed by a seven-day food diary. There was dose-dependent small bowel damage: none of the four children on strict GFD had mucosal abnormality while all six of children who were consuming up to 500 mg of gluten daily had severe villous atrophy. Serological tests were not helpful in identifying the mucosal changes.

In a study conducted in Finland, Collin et al. [28] used four-day food records to estimate the daily gluten intake in 76 adults and 16 children with CD managed with a strict GFD for a median of 2 years. Twenty-eight subjects were eating naturally gluten-free products while 48 consumed wheat starch-based gluten-free products. The gluten content of various products was assessed: five of the naturally gluten-free and two of the wheat starch-based products contained more than 100 parts per million (ppm) of gluten. However, the authors did not demonstrate any correlation between the use of the products and mucosal histology. In the children, despite ingestion of up 140 g of products daily, none of the children had abnormal small bowel biopsies.

In a different approach assessing short-term exposure, Ciclitira et al. [29] showed that 10 mg gliadin administered directly by intraduodenal infusion over 8 h did not lead to jejunal changes in one patient with CD, while administration of 100 mg gliadin produced minor changes. These investigators also showed significant changes (reduction in mean VH/CrD ratio) in seven patients with CD who were given between 1.2 and 2.4 mg gliadin a day in gluten-free bread in addition to their GFD [30]. Surprisingly, however, they did not show any significant difference in jejunal morphometry during a 6-week period of ingestion of 1.2 mg and 2.4 mg gluten in bread compared to a separate similar period without consumption of bread gluten [31].

Kaukinen et al. [32] did not show any relationship between the daily gluten intake (mean 35 mg gluten daily: range between 5 and 150 mg daily) and mucosal structure (assessed as VH/CrD and enterocyte height) in 52 subjects with CD. None of these patients were shown to have positive EMA or anti-reticulin antibodies. Three had positive AGA antibodies despite normal histology. Similarly, Lohiniemi et al. [33] also reported a lack of symptoms in a group of 58 adults with CD consuming an average of 36 mg of gluten (range 0–180 mg) daily. Twenty-one of 23 subjects had normal

small bowel biopsies. Just one patient had subtotal villous atrophy, while one subject had partial villous atrophy. On the other hand, a smaller amount of gluten (1.5 mg daily) administered for 12 months resulted in ongoing symptoms in 11 of 17 subjects and intermittent symptoms in five patients [22]. This gluten exposure did not lead to changes in standard AGA tests. Low titres of AGA were seen in a group who continued this exposure for longer (a mean of 6 years). As small bowel biopsies were not obtained in this study, the full relevance of this exposure was not able to be fully assessed.

A further study also assessed gluten exposure over a similar time period [34]. Twenty-four children with proven CD were asked to take 10 g of gluten daily for up to 51 weeks. Although the actual intake was recorded at only 20–260 mg daily, there was no relationship between intake of gluten and symptoms. Of the 23 children who had their small bowel biopsy re-assessed after the challenge, 22 had an increased number of intraepithelial lymphocytes, which was dependent upon the dose of gluten taken. In addition, specific antibodies were detected in most of these children after 2 months of the challenge.

A dose-dependent relationship between gluten exposure and mucosal damage was seen in a further prospective study involving 123 adolescents with CD [35]. The children were classified as being on a strict GFD, those having occasional intake (averaging 0.73 g/day) and others having a gluten-containing diet. All the children on strict GFD had normal small bowel histology. Most of the 14 children with less strict GFDs had abnormalities ranging from villous shortening to villous atrophy. Neither symptoms nor serological testing were reliable indicators in this group. Marked histological changes were detected in all but one of the children on a gluten-containing diet. In contrast, Biagi et al. [19] studied the effect of consuming a small amount of gluten over 24 months in a woman with CD. They showed that the ingestion of just 1 mg of gluten per day prevented histological recovery.

Some of these studies raise the ethical question of exposing patients with CD to gluten. Nevertheless, all these studies were completed in the era between the 1990 and 2011 European Society for Pediatric Gastroenterology, Hepatology, and Nutrition (ESPGHAN) guidelines when gluten challenge was part of the protocol for the diagnosis of CD [36,37].

**Table 1.** Relationships between gluten exposure and outcome.

| Reference | Population | Gluten Amount | Outcome | Remarks |
|---|---|---|---|---|
| Jansson 2001, [23] | 54 children | 0.2/0.5 g/kg | All relapsed within 8 weeks. Higher dose relapsed earlier | The severity of mucosal changes was greater in those who had received 5 g/kg/day |
| Catassi 1993, [24] | 20 children | 100 or 500 mg/day for 4 weeks | Those who received 100 mg had minimal histological changes. The children in the higher dose group had marked mucosal damage | |
| Catassi 2007, [25] | 49 adults | 10 mg of gluten daily, 50 mg of gluten daily or placebo | The subjects given the higher dose of gluten for three months had significant mucosal damage 7/13 subjects in the low-dose gluten group also had worsening of their VH/CrD ratio | No significant change in IgA anti-tissue transglutaminase antibodies was noted |
| Selby 1999, [26] | 89 adults | 39 received what the authors called a Codex GFD (allows up to 0.03% protein derived from gluten-containing grains and 50 patients consuming a non-gluten detectable-GFD | The presence of intra-epithelial lymphocytes or villous atrophy did not differ between groups | |
| Troncone 1995, [27] | 10 adolescents | None (4 children) Up to 500 mg daily (6 children) | None of the four children on strict GFD had mucosal abnormality while all six of children who were consuming up to 500 mg of gluten daily had severe villous atrophy | Seven-day food diary. Serological tests were not helpful in identifying the mucosal changes |
| Collin 2004, [28] | 76 adults and 16 children | Twenty-eight subjects were eating naturally gluten-free products while 48 consumed wheat starch-based gluten-free products | No correlation between the use of the products and mucosal histology. None of the children had abnormal small bowel biopsies | |
| Ciclitira 1984, [30] | 7 adults | 1.2 and 2.4 mg gliadin a day in gluten-free bread in addition to their GFD | Significant reduction in mean VH/CrD ratio | |

**Table 1.** *Cont.*

| Reference | Population | Gluten Amount | Outcome | Remarks |
|---|---|---|---|---|
| Kaukinen 1999, [32] | 41 children and adults with CD and 11 adults with dermatitis herpetiformis | Mean consumption 34 mg (5–150 mg) daily gluten intake | No relationship between the daily gluten intake and mucosal structure | |
| Lohiniemi 2000, [33] | 58 adults | An average of 36 mg of gluten (range 0–180 mg) daily gluten intake | Twenty-one of 23 subjects had normal small bowel biopsies. Just one patient had subtotal and while one subject had partial villous atrophy | No symptoms noted |
| Laurin 2002, [34] | 24 children | 10 g of gluten daily for up to 51 weeks. Actual intake only 20–260 mg daily | Of the 23 children who had their small bowel biopsy re-assessed after the challenge, 22 had an increased number of intraepithelial lymphocytes (dependent upon the gluten dose) | No relationship between gluten intake and symptoms. Specific antibodies were detected in most of these children after 2 months of the challenge |
| Mayer 1991, [35] | 123 adolescents | The children were classified as being on a strict GFD, those having occasional intake (averaging 0.73 g/day) and others having a gluten-containing diet | All the children on strict GFD had normal small bowel histology. Most of the 14 children with less strict GFD had abnormalities ranging from villous shortening to villous atrophy. Marked histological changes were detected in all but one of the children on a gluten-containing diet | Neither symptoms nor serological testing were reliable indicators in this group |

CD, celiac disease; GFD, gluten-free diet; VH/CrD, villous height/crypt depth.

## 4. Safe Gluten Exposure Limits Arising from Review Articles

The amount of daily gluten exposure within a GFD that could be considered to be safe has been discussed in several review papers. Firstly, Hischenhuber et al. [38] suggested patients with CD should have between 10 and 100 mg of gluten daily. On the other hand, Akobeng et al. [39] suggested that daily gluten intake of <10 mg is unlikely to cause significant histological abnormalities.

The Food and Drug Administration (FDA) of the USA issued a health hazard assessment for gluten exposure in individuals with CD in 2011 [40]. Based on a comprehensive review of the literature available at that time, this report concluded that the safe tolerable amount of gluten is 7 mg daily to avoid histological changes and 0.015 mg each day to prevent symptoms.

## 5. What Should Be Labeled as a Gluten-Free Product?

Gibert et al. [41] collected data on the intake of gluten-free products by patients with CD in two Mediterranean countries (Italy and Spain) and two northern European countries (Norway and Germany). The authors suggested that gluten-free products should be defined as having a gluten content of less than 20 ppm. In contrast, Dessi et al. [9] concluded that wheat starch-based foods are safe, providing that they contain less than 100 mg of gluten/kg.

The Codex Alimentarius, which represents a consensus of international standards for food safety, established 20 ppm as the cutoff for products to be called gluten-free [42]. The updated European Union (EU) legislation, published in 2014, specifies two subgroups: gluten-free (≤20ppm) and very low gluten (≤100 ppm) [43]. Although many countries (for example Spain, Italy, UK, Canada and US) [42] use the Codex definition of 20 ppm, others use different cutoffs. These include 10 ppm in Argentina and 3 ppm (detection limit) in Australia, New Zealand and Chile [8]. Given the increased ability to detect trace elements of gluten when lower than 20 ppm, the definition of this being gluten-free could be seen as a misnomer.

According to the FDA Docket No. FDA-2005-N-0404 [42], approximately 5% of foods labeled "gluten-free" in the USA do not meet the definition for "gluten-free", putting individuals diagnosed with CD at risk. Therefore, the FDA prohibit the use of the "gluten-free" label in foods that contain 20 or more parts per million (ppm) gluten, and also prohibit the gluten-free claim on foods that have any quantity of certain ingredients. If a food contains wheat, rye, barley, or their crossbred hybrids, it cannot be labeled "gluten-free". They state that currently there is no reliable test method that can quantify the amount of gluten in foods that have been fermented or hydrolyzed.

This issue raises the prospect of legal consequences for companies that use the label "gluten free" as a label when this is not strictly so. This, in turn, highlights the need for health authorities to demand the actual gluten content in each gluten-free product is stated.

## 6. Conclusions

At present, there is no clear consensus on the safe amount of daily gluten intake for individuals with CD on a GFD. Some of the variation in evidence reflects differences between individuals, the methods of assessing gluten intake and marked differences in methodology between the reported studies. For instance, food diary records may under or overestimate actual intake.

We would like to emphasize the importance of reading labels and appropriate interpretation of their information. For most patients with CD, a daily consumption of less than 50 mg of gluten appears to be safe. However, mucosal changes can be triggered in some individuals with as little as 10 mg of gluten daily. In the absence of definitive data and until further studies provide more clarity, children and adults are recommended to have a GFD with foods containing less than 20ppm as defined by the Codex. Patients should be informed to avoid excessive consumption of this kind of product, since gluten could accumulate up to an unacceptable limit. It is also important to be aware that some CD patients will not reveal any symptom but will develop mucosal damage if exposed.

**Funding:** This review received no external funding.

**Conflicts of Interest:** The authors declare no conflict of interest.

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
