# Peer review of "Should the Glu Be Ten or Twenty? An Update on the Ongoing Debate on Gluten Safety Limits for Patients with Celiac Disease"

_gastrointestdisord, doi:10.3390/gidisord2030021_

Round 1

Reviewer 1 Report

REVIEW PROCESS gastrointestinal disorders

MAJOR COMMENTS

There is little novelty in this review, as the studies you mentioned are reviewed in previous works such that of Akobeng (2008) you cited in reference 37. I suggest to shorten the review of clinical studies and go further into regulations, labels and last decisions about how to inform celiac consumer or patient about the real risk they take.

L 23-27. There are data available for CD prevalence in Europe and other parts of the world. Some suggestions:

  • Lionetti E, Catassi C. New clues in celiac disease epidemiology, pathogenesis, clinical manifestations, and treatment. Int Rev Immunol. 2011;30(4):219-31.
  • Nevoral J. Celiac Disease in Children: What Has Changed? Int J Cel Dis. 2014;2(1):18-23.
  • Peña A, Rodrigo L. Epidemiology of Celiac Disease and Non-Celiac Gluten-Related Disorders. In: Arranz E, Fernández-Bañares F, Rosell C, Rodrigo L, Peña A, editors. Advances in the Understanding of Gluten Related Pathology and the Evolution of Gluten-Free Foods. Barcelona, Spain: OmniaScience; 2015. p. 27-73.
  • Catassi C, Rätsch IM, Gandolfi L, Pratesi R, Fabiani E, El Asmar R, et al. Why is coeliac disease endemic in the people of the Sahara? Lancet. 1999;354(9179):647-8.
  • Lionetti E, Gatti S, Pulvirenti A, Catassi C. Celiac disease from a global perspective. Best Pract Res Clin Rheumatol. 2015;29(3):365-79.
  • Catassi, C.; Gatti, S.; Fasano, A. The new epidemiology of celiac disease. J Pediatr Gastroenterol Nutr 2014, 59 Suppl 1, S7-9, doi:10.1097/01.mpg.0000450393.23156.59.

L 59. Since gluten is not totally absent in the so-called “gluten-free products”, the important issue then is to define the safe gluten amount for individuals with CD[9].

This sentence is misunderstanding. Gluten-free products can be totally absent of gluten, don´t they?. Especially if, as you told, they are completely formulated with gluten-free cereals or ingredients. This is much more complex, as it involves cross-contamination processes and labelling is dependent on each country´s legislation.

My suggestion, to include the following aspects in red: The standard GFD prescription involves the use of foods that are naturally gluten-free (such as vegetables, legumes, naturally gluten-free cereals and fruits). Even if some products, especially processed ones, are labeled as “gluten-free”, gluten is not always totally absent, as they could contain up to 20 ppm to bear such a claim, depending on the legal status of each country. Consequently, it is important to define the safe gluten amount for individuals with CD and to inform patients about food labels, in order to limit gluten ingestion.

L 102. This paragraph is misunderstanding. A Codex-GFD doesn´t exist. It is an expression used by the authors of the referenced paper. What the Codex Alimentarius has published is a standard defining the conditions of gluten-free products, which you mentioned further on, but not diet. Please, correct it in the text and in the table and better explain the experiment. Nevertheless, this is not the best study to be discussed, as they did not measured gluten intake, and it is not possible to relate to a quantifiable ingestion limit.

L215. Why did you reflect only about children in your conclusion?. In fact, the review is not focused on this collective. Moreover, I think you should go further, and send a message to highlight the importance of reading labels and appropriately interpret their information. You recommend GFD and products containing < 20 ppm of gluten, but you should prevent patients about excessive consumption of this kind of products, because 5 mg per day is acceptable, but people consuming processed products as an important part of their diet could accumulate gluten up to an unacceptable limit. Considering that some of them will not reveal any symptom but will develop mucosal damage, it is important to be aware.

MINOR COMMENTS

L28. Cereals containing gluten are more than wheat, rye and barley, even if those ones are the most used ones. I suggest to add “such as” to rye and barley.

L 87. Please correct, 5 g/kg/day à 0,5 g/kg/day

Table1. Could you please specify the year of the study in the table? It could be of interest.

L161. This reference does not show the gluten dose administered. Its objective is not to set an ingestion threshold, but to determine if IgA-AGA could be used for CD diagnosis, instead of a biopsy, in children. I do not think that it is relevant for the purpose of this review. 

L41. Please, provide the legal reference for this sentence, and not an article. Moreover, EU legislation regarding gluten free product´s label was updated in 2014 (COMMISSION IMPLEMENTING REGULATION (EU) No 828/2014 of 30 July 2014, on the requirements for the provision of information to consumers on the absence or reduced presence of gluten in food).

L195. May be it is useful to highlight that the cutoff of 3 ppm is due to the detection limit of the procedure?

L203. Please, add to this sentence the adjective “reliable” or something similar. In fact, AOAC and AACC have recognized a method based on polyclonal antibodies (AOAC (2015.05 method) and AACC (38-55.01 method).  This method is the one recommended for that kind of fermented or hydrolyzed food.

L206. I suggest adding a reflection about the fact that reflecting exactly the actual gluten content will increase the final price of those expensive products.

Reviewer 2 Report

The review by Cohen et al, intends to update on the debate on the safe amount of daily gluten intake for individuals with CD on a GFD. The authors consider in the first part of the review very old literature, in which are considered studies with histological and serological parameters that have been evolved along the time, for this reason, It could be a good idea to cite the studies in a chronological order to highlight how the question about daily gluten intake has evolved during the time, otherwise, it is confusing.

Some time is difficult to follow, some of the sentences describing the studies are not always clear.

Specific comments

Line 15:   not universally effective or tolerated, moderate this sentence, GFD has not to be tolerated, but strict adherence to diet is not easy to accomplish

Line 17: it could be useful to specify what parts per million means ( mg/Kg)

Line 25 CD is 2present in around 1-2% of populations in North and South America, North Africa, Middle East, and India add also Oceania. CD is present quite in all the world, simply there are not reporting data or there is a higher prevalence in some populations ( North Europe, Sahrawi ), please consider this reference Singh P, Arora A, Strand TA, et al. Global Prevalence of Celiac Disease: Systematic Review and Meta-analysis. Clin Gastroenterol Hepatol. 2018;16(6):823‐836.e2.

Line 29 ….10–20 grams per person per day   Add a reference for this statement

Line 35 these residues present in the undigested gluten fragments contribute to the deamination of the peptides by tissue 35 transglutaminase (tTG) within the lamina propria… change the sentence, residues of prolamine and glutamine do not contribute to deamidation itself, but are excellent substrates for tTG.

Line 36 please do not use the word deleterious is too much! The sentence must be clarified, suggesting the presence of toxic sequences for the observed effect ( where? In which model?)

Line 41 which T lymphocytes?

Line n54 change ongoing with other words

Line 58 , not only vegetables and fruits!

Line 76 specify the study

Line 79 The presence of persisting changes was particularly associated with dietary adherence. This sentence is not clear

Line 82-88 The study it is not clearly described. Do the patients were divided in different groups exposed to different gluten doses ( 0,2, 0,5, and 5 g/Kg day)?

Line 100 Allowing for the 99 morphological changes, what do the authors mean?

Line 126   The same fashion …Do the authors mean similarly, in the same way?

Table 1 Year of the reference is missing, include a column of the parameter/analysis investigated

Line 195-197 The concept it is not clear

Line 213 Cite the paper that state this affirmation ( 50 mg)

Line 215 Children but also adults are supposed

Reference n 14, pages are missing

Author Response

Reviewer 2

Comments and Suggestions for Authors

The review by Cohen et al, intends to update on the debate on the safe amount of daily gluten intake for individuals with CD on a GFD. The authors consider in the first part of the review very old literature, in which are considered studies with histological and serological parameters that have been evolved along the time, for this reason, It could be a good idea to cite the studies in a chronological order to highlight how the question about daily gluten intake has evolved during the time, otherwise, it is confusing.

Some time is difficult to follow, some of the sentences describing the studies are not always clear.

Specific comments

Line 15:   not universally effective or tolerated, moderate this sentence, GFD has not to be tolerated, but strict adherence to diet is not easy to accomplish

The sentence has been changed as suggested.

Line 17: it could be useful to specify what parts per million means ( mg/Kg)

This has been added as suggested

Line 25 CD is 2present in around 1-2% of populations in North and South America, North Africa, Middle East, and India add also Oceania. CD is present quite in all the world, simply there are not reporting data or there is a higher prevalence in some populations ( North Europe, Sahrawi ), please consider this reference Singh P, Arora A, Strand TA, et al. Global Prevalence of Celiac Disease: Systematic Review and Meta-analysis. Clin Gastroenterol Hepatol. 2018;16(6):823‐836.e2.

The reference has been added as suggested

Line 29 ….10–20 grams per person per day   Add a reference for this statement. The reference has been added

Line 35 these residues present in the undigested gluten fragments contribute to the deamination of the peptides by tissue 35 transglutaminase (tTG) within the lamina propria… change the sentence, residues of prolamine and glutamine do not contribute to deamidation itself, but are excellent substrates for tTG.

Sentence has been changed as suggested

Line 36 please do not use the word deleterious is too much! The sentence must be clarified, suggesting the presence of toxic sequences for the observed effect ( where? In which model?)

Sentence has been changed to: Of the four described fractions of gliadin, the α-gliadin subunit, due to its high T-cell–stimulatory sequences has the most  significant effects while β, γ and ω subunits having milder toxicity

Line 41 which T lymphocytes?

We added “some subsets of CD4 and CD8

Line n54 change ongoing with other words

Ongoing changed to continuous

Line 58 , not only vegetables and fruits!

The sentence has been changed to “It is composed of naturally gluten-free foods and certified processed gluten-free products”

Line 76 specify the study

We expanded the data on the study

Line 79 The presence of persisting changes was particularly associated with dietary adherence. This sentence is not clear

Persisting changes was changed to intestinal damage

Line 82-88 The study it is not clearly described. Do the patients were divided in different groups exposed to different gluten doses ( 0,2, 0,5, and 5 g/Kg day)?

This was a mistake only 0.2 and 0.5g/Kg. It is corrected now.

Line 100 Allowing for the 99 morphological changes, what do the authors mean?

We replaced it to Interestingly

Line 126   The same fashion …Do the authors mean similarly, in the same way?

We changed the sentence to “while administration of 100 mg gliadin produced minor changes.”

Table 1 Year of the reference is missing, include a column of the parameter/analysis investigated

This has been added

Line 195-197 The concept it is not clear

Line 213 Cite the paper that state this affirmation ( 50 mg)

Line 215 Children but also adults are supposed

Reference n 14, pages are missing

Round 2

Reviewer 1 Report

The authors changed important issues in the manuscript, as suggested, and now it is precise enough to be published. Nevertheless, some minor aspects and one major aspect that should be considered in the previous round must be reconsidered:

L64 of the new version. The paragraph is now repeating information from paragraph in line 51. It sounds repetitive.

L190. Of the new version: The European Union (EU) legislation, published in 2009 and regulated in 2012, specifies two subgroups: gluten-free (≤20ppm) and low gluten (21-100 ppm)[43].

Please, provide the legal reference for this sentence, and not an article. Moreover, EU legislation regarding gluten free product´s label was updated in 2014 (COMMISSION IMPLEMENTING REGULATION (EU) No 828/2014 of 30 July 2014, on the requirements for the provision of information to consumers on the absence or reduced presence of gluten in food).

Nevertheless, the classification of food products remained the same.

L211 of the new version. Please check this sentence, it is strange.

Author Response

The authors changed important issues in the manuscript, as suggested, and now it is precise enough to be published. Nevertheless, some minor aspects and one major aspect that should be considered in the previous round must be reconsidered:

L64 of the new version. The paragraph is now repeating information from paragraph in line 51. It sounds repetitive.

We thank the reviewer for this comment and the repeated sentence was deleted.

L190. Of the new version: The European Union (EU) legislation, published in 2009 and regulated in 2012, specifies two subgroups: gluten-free (≤20ppm) and low gluten (21-100 ppm)[43].

Please, provide the legal reference for this sentence, and not an article. Moreover, EU legislation regarding gluten free product´s label was updated in 2014 (COMMISSION IMPLEMENTING REGULATION (EU) No 828/2014 of 30 July 2014, on the requirements for the provision of information to consumers on the absence or reduced presence of gluten in food). Nevertheless, the classification of food products remained the same.

We thank the reviewer for this and the legal reference has been added instead of the paper and the text was updated accordingly.

L211 of the new version. Please check this sentence, it is strange.

This sentence was added as part of our understanding of the reviewers’ suggestions:

The sentence refers to the importance of labels reading by the consumers. We have modified the sentence:  to “We would like to emphasize the importance of reading labels and appropriately interpretation of their information. Nevertheless, we can delete the sentence if the reviewer finds it unnecessary.

Reviewer 2 Report

The authors answered to the questions requested

Author Response

We thank the reviewer.